

# The prognostic and clinicopathological significance of SLC7A11 in human cancers: a systematic review and meta-analysis

Jiantao Wang[1,*], Siyuan Hao[2,*], Guojiao Song[1], Yan Wang[2] and Qiukui Hao[3]

[1] Sichuan University, State Key Laboratory of Biotherapy & Department of Lung Cancer Center and Department of Radiation Oncology, West China Hospital, Chengdu, China
[2] Sichuan University, State Key Laboratory of Oral Diseases & National Clinical Research Center for Oral Diseases & Department of Pediatric Dentistry, West China Hospital of Stomatology, Chengdu, China
[3] McMaster University, Faculty of Health Sciences, School of Rehabilitation Science, Ontario, Canada
[*] These authors contributed equally to this work.

## ABSTRACT

**Objective**. It is of great importance to recognize bio-markers for cancer prognosis. However, the association between solute carrier family 7 member 11 (SLC7A11) and prognosis is still controversial. Therefore, we conducted this systematic review and meta-analysis to identify the prognostic and clinicopathological significance of SLC7A11 in human cancers.

**Methods**. PubMed, Web of Science, Scopus, the Cochrane Library and Embase database were searched from database inceptions to March 19th 2022. Hand searches were also conducted in references. Prognosis and clinicopathological data were extracted and analyzed.

**Results**. A total of 12 eligible studies with 1,955 patients were included. The results indicated that SLC7A11 expression is associated with unfavorable overall survival (OS), unfavorable recurrence-free survival (RFS) and unfavorable progression free survival (PFS). And SLC7A11 expression is also associated with more advanced tumor stage.

**Conclusions**. SLC7A11 expression is associated with more unfavorable prognosis and more advanced tumor stage. Therefore, SLC7A11 could be a potential biomarker for human cancer prognosis.

# INTRODUCTION

Cancer is the leading cause of death worldwide, and has been a severe public health burden with increased mortality and morbidity. Thus, it is of great importance to identify biomarkers for early diagnosis, prognosis assessment and precision treatment for cancer patients (*Siegel et al., 2021*). Therefore, SLC7A11 could be a potential biomarker for human cancer prognosis.

The solute carrier (SLC) series includes more than 60 gene families with a total of more than 400 members, encoding most human transporters, of which solute carrier family 7

Corresponding author
Jiantao Wang,
wangjiantao@wchscu.cn

(SLC7) is mainly involved in the transport of amino acids on the plasma membrane (*Lin et al., 2020*). The SLC7A11 gene is located on human chromosome 4 and contains 14 exons, and is widely expressed in tissues and cells such as brain, liver, macrophages and retinal pigment cells (*Hong et al., 2021*). The SLC7A11 gene encodes SLC7A11 (also known as xCT) protein, which acts as a light chain subunit and forms the $Xc^-$ cystine/glutamate antiporter with the heavy chain subunit SLC3A2 (also known as 4F2hc) (*Koppula et al., 2018*). As an antiporter, SLC7A11 mediates the intake of cystine and the exchange of glutamate (*Luo et al., 2022*). After taken into the cell, cystine would be reduced into cysteine, which plays an important role in protecting cells from oxidative stress (*Shao et al., 2022*). SLC7A11 has been observed to express in various cancer tissues and plays different functions in a number of pathophysiological processes, including ferroptosis, immune system function, metabolic flexibility/nutrient dependency and redox homeostasis (*Jyotsana, Ta & DelGiorno, 2022*). For example, tumor cells can maintain high levels of glutathione by upregulating the expression of the catalytic subunit SLC7A11 of the $Xc^-$ system to counteract the oxidative stress caused by the increased rate of their own metabolism (*Koppula, Zhuang & Gan, 2021*). *Gout et al. (2001)*'s research pointed out that the growth of lymphocytes depends on the uptake of cystine/cysteine from the microenvironment by systems such as $Xc^-$, and the ability to increase cystine uptake is a potential process for the progression of T-cell malignancies. The activity of the $Xc^-$ system and intracellular glutathione levels are closely related to lymphoma cell growth (*Guan et al., 2009*). By constructing SLC7A11 knockout in U251 glioma cells, Polewski et al. (*2017*; *2016*) found that SLC7A11 knockout could increase basal reactive oxygen species levels, reduce glutathione generation, and promote oxidative stress and genotoxic stress. SLC7A11 overexpression increases glioma cell resistance to oxidative stress and reduces sensitivity to temozolomide (*Polewski et al., 2017*; *Polewski et al., 2016*).

However, the association between solute carrier family 7 member 11 (SLC7A11) and prognosis is still controversial. Therefore, we conducted this systematic review and meta-analysis to identify the prognostic and clinicopathological significance of SLC7A11 in human cancers.

## METHODS

The protocol of this systematic review and meta-analysis has been registered on PROSPERO website (http://www.crd.york.ac.uk/PROSPERO). The ID number is CRD42022321708. We followed the Preferred Reporting Items for Systematic Reviews and Meta-Analyses (PRISMA) (*Moher et al., 2009*). And we referred to the methods used in our previously published paper (*Wang et al., 2022*).

### Search strategy

PubMed, Web of Science, Scopus, the Cochrane Library and Embase database were searched. We used key words including SLC7A11, cancer, tumor, neoplasms, and prognosis. The strategy of literature search was shown in Table 1.

**Table 1** **Our search strategy.** PubMed, Web of Science, Scopus, the Cochrane Library and the Embase database were searched, from the establishment of these databases to March 19th 2022.

| Electronic database and search strategy | |
| --- | --- |
| PubMed | #1 ((((((((((((((Neoplasms[MeSH Terms]) OR (Neoplasia[Title/Abstract])) OR (Neoplasias[Title/Abstract])) OR (Neoplasm[Title/Abstract])) OR (Tumors[Title/Abstract])) OR (Tumor[Title/Abstract])) OR (Cancer[Title/Abstract])) OR (Cancers[Title/Abstract])) OR (Malignancy[Title/Abstract])) OR (Malignancies[Title/Abstract])) OR (Malignant Neoplasms[Title/Abstract])) OR (Malignant Neoplasm[Title/Abstract])) OR (Neoplasm, Malignant[Title/Abstract])) OR (Neoplasms, Malignant[Title/Abstract])) |
| | #2 ((((Solute Carrier Family 7 Member 11[Title]) OR (SLC7A11[Title])) OR (XCT[Title])) OR (CCBR1[Title])) OR (system Xc-[Title]) |
| | #3 #1 AND #2 |
| Cochrane Library | #1 MeSH descriptor: [Neoplasm] explode all trees |
| | #2 (Solute Carrier Family 7 Member 11):ti,ab,kw |
| | #3 (SLC7A11):ti,ab,kw |
| | #4 (XCT ):ti,ab,kw |
| | #5 (system Xc-):ti,ab,kw |
| | #6 (CCBR1):ti,ab,kw |
| | #7 #2 OR #3 OR #4 OR #5 OR #6 |
| | #8 #1 AND #7 |
| Embase | #1 'malignant neoplasm'/exp |
| | #2 'tumor':ab,ti |
| | #3 'neoplasm':ab,ti |
| | #4 'cancer':ab,ti |
| | #5 #1 OR #2 OR #3 OR #4 |
| | #6 'solute carrier family 7 member 11'/exp |
| | #7 'solute carrier family 7 member 11 protein'/exp |
| | #8 'slc7a11':ti |
| | #9 'xct':ti |
| | #10 'system xc-':ti |
| | #11 'ccbr1':ti |
| | #12 #6 OR #7 OR #8 OR #9 OR #10 #11 |
| | #13 #5 AND #12 |
| Web of Science | (AB=(Neoplasms OR Tumor OR Cancer OR Malignancy OR Malignant Neoplasm) AND TI=(Solute Carrier Family 7 Member 11 OR SLC7A11 OR XCT OR system Xc- OR CCBR1)) |
| | Indexes=SCI-EXPANDED, SSCI, A&HCI, CPCI-S, CPCI-SSH, ESCI, CCR-EXPANDED, IC Timespan=All years |

**Table 1** (*continued*)

**Electronic database and search strategy**

| Scopus | (TITLE-ABS-KEY ('malignant neoplasm') OR TITLE-ABS-KEY ('tumor') OR TITLE-ABS-KEY ('neoplasm') OR TITLE-ABS-KEY ('carcinoma') OR TITLE-ABS-KEY ('cancer')) AND (TITLE('Solute Carrier Family 7 Member 11') OR TITLE('SLC7A11') OR TITLE('XCT') OR TITLE('system Xc-') OR TITLE('CCBR1')) |
|---|---|

## Inclusion and exclusion criteria

The inclusion criteria were as followings: (1) Focusing on the association between SLC7A11 and prognosis. (2) Subjects should be diagnosed by pathology. (3) SLC7A11 expression was evaluated by real-time quantitative polymerase chain reaction (RT-qPCR) or immunohistochemistry (IHC). (4) The survival and clinicopathological data should be directly provided or could be extracted. The exclusion criteria were as followings: (1) The survival and clinicopathological data could not be extracted. (2) The survival and clinicopathological data could not be retrieved by contacting the authors. (3) *In vitro* studies, bioinformatics analyses, case-reports, conferences abstracts or reviews. (4) The language of the study was not English.

## Quality assessment of included studies

Newcastle-Ottawa Scale (NOS) was utilized to evaluate the studies' quality. A score of six and more were recognized as high quality study. The assessment was conducted by two authors independently. Disagreements were settled by discussion with other authors.

## Data extraction

We extracted the following features from included studies: author's name, published year, type of cancer, sample size, assessment of SLC7A11 expression and cut-off value. If the HR was not provided directly, we would extracted the HR according to the method published by *Tierney et al. (2007)*. Firstly, we extracted the survival curve plot from the article. Secondly, the Engauge Digitizer software was used to extract the survival rate and time data from the curve. Thirdly, The HR was calculated in Excel software. Clinicopathological data were also extracted. If the data could not be extracted directly, we would contact the authors of included studies.

## Statistical analysis

The combined HR and 95% confidence interval (CI) were utilized to assess the association between SLC7A11 and prognosis. If $P < 0.05$ and HR $>1$, it suggested that SLC7A11 expression was associated with unfavorable prognosis. If $P < 0.05$ and HR $<1$, it suggested that SLC7A11 expression was associated with better prognosis. If $P > 0.05$, it suggested that SLC7A11 expression was not significantly associated with prognosis. The combined OR and 95% CI were utilized to assess the association between SLC7A11 and clinicopathological parameters. Cochrane's I-squared statistic ($I^2$) was applied to evaluate the heterogeneity. $I^2 > 50\%$ was regarded as significant heterogeneity and random effect model would be conducted. Otherwise, a fixed effect model would be used. All these analyses were performed in R 4.2.1 (*R Core Team, 2022*) with the "meta" package.

## RESULTS

### Search results

Firstly, 220 studies were recognized in PubMed, 242 in Web of Science, 261 in Scopus, one in the Cochrane Library and 322 in Embase database. After excluding duplications, 401 articles remained. Secondly, title sand abstracts were read and 376 articles were excluded sense they were no related to topic. Thirdly, the remaining 25 articles were read in detail. Two case reports, seven conference abstracts, four bioinformatics analyses were excluded. Ultimately, 12 studies containing 1,955 patients were selected for systematic review and meta-analysis (*Feng et al., 2021*; *Ji et al., 2018*; *Kinoshita et al., 2013*; *Lee et al., 2018*; *Ma et al., 2017*; *Namikawa et al., 2015*; *Shiozaki et al., 2014*; *Sorensen et al., 2018*; *Sugano et al., 2015*; *Takeuchi et al., 2013*; *Toyoda et al., 2014*; *Zhang et al., 2018a*) (Fig. 1).

### Study characteristics

The characteristics of included studies were exhibited in Table 2. There were 10 types of cancer, including esophageal squamous cell carcinoma, non-small cell lung cancer, hepatocellular carcinoma, oral cavity squamous cell carcinoma, laryngeal squamous cell carcinoma, glioma, colorectal cancer, glioblastoma, tongue cancer, and liver carcinoma. SLC7A11 expression was assessed by RT-qPCR or IHC. The follow-up duration and outcomes were also exhibited.

### Quality assessment

The scores of NOS were exhibited in Table 3. With the mean score of 7.25, the total quality was recognized as relatively high. In detail, two studies had a score of 6 (*Kinoshita et al., 2013*; *Lee et al., 2018*), five achieved a score of 7 (*Ji et al., 2018*; *Sorensen et al., 2018*; *Sugano et al., 2015*; *Toyoda et al., 2014*; *Zhang et al., 2018a*) and five had a score of eight (*Feng et al., 2021*; *Ma et al., 2017*; *Namikawa et al., 2015*; *Shiozaki et al., 2014*; *Takeuchi et al., 2013*).

### Correlation between SLC7A11 expression and OS

Ten studies reported the correlation between SLC7A11 expression and OS (*Feng et al., 2021*; *Ji et al., 2018*; *Kinoshita et al., 2013*; *Lee et al., 2018*; *Ma et al., 2017*; *Shiozaki et al., 2014*; *Sorensen et al., 2018*; *Takeuchi et al., 2013*; *Toyoda et al., 2014*; *Zhang et al., 2018a*). The pooled HR showed that the high expression of SLC7A11 was statistically associated with unfavorable univariate OS (HR $=1.93$, 95% CI [1.60–2.30], $I^2 = 0\%$, fixed effect model) (Fig. 2A). The multivariate OS was also combined (*Feng et al., 2021*; *Ji et al., 2018*; *Kinoshita et al., 2013*; *Lee et al., 2018*; *Shiozaki et al., 2014*; *Zhang et al., 2018a*). The results indicated that high expression of SLC7A11 was significantly associated with poor multivariate OS (HR $= 1.63$, 95% CI [1.29–2.05], $I^2 = 35\%$, random effect model) (Fig. 2B).

### Correlation between SLC7A11 expression and RFS

Four studies reported the correlation between SLC7A11 expression and RFS (*Kinoshita et al., 2013*; *Lee et al., 2018*; *Ma et al., 2017*; *Sugano et al., 2015*). The pooled HR suggested that the high expression of SLC7A11 was statistically associated with unfavorable univariate RFS (HR $= 2.01$, 95% CI: [1.57–2.57], $I2 = 0\%$, fixed-effect model) (Fig. 2C). In the meanwhile, the combined multivariate RFS also showed that high expression of SLC7A11

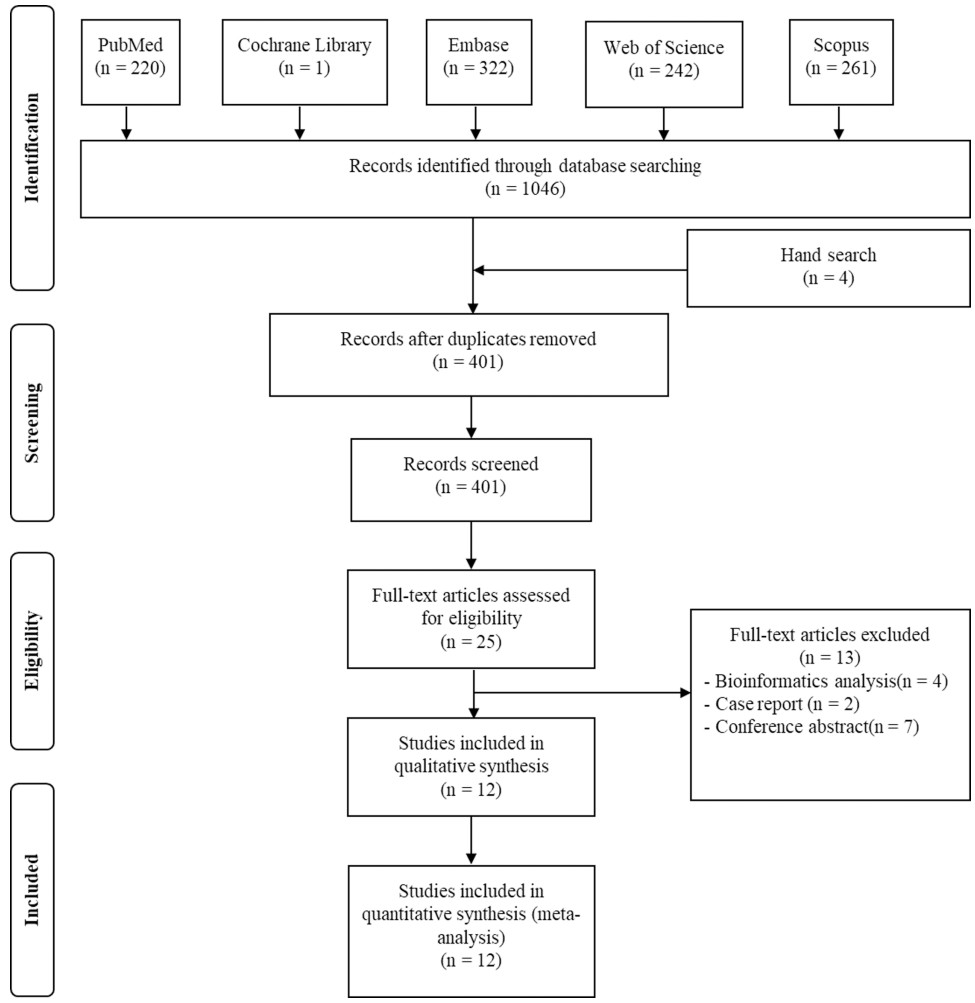

**Figure 1  Flow plot exhibiting the literature search.** Flow plot exhibiting the literature search in PubMed, Web of Science, Scopus, the Cochrane Library and the Embase database, from the establishment of these databases to March 19th 2022.

was statistically correlated with unfavorable RFS (HR = 1.99, 95% CI [1.45–2.74], I2 = 0%, fixed-effect model) (Fig. 2D).

## Correlation between SLC7A11 expression and PFS

Two studies reported the correlation between SLC7A11 expression and PFS (*Takeuchi et al., 2013*; *Toyoda et al., 2014*). The pooled HR suggested that the high expression of SLC7A11 was statistically associated with unfavorable univariate PFS (HR = 2.61, 95% CI [1.22–5.57], I2 = 49%, fixed-effect model) (Fig. 2F). Only *Takeuchi et al. (2013)* reported the multivariate PFS and showed that high expression of SLC7A11 was statistically correlated with unfavorable PFS (HR = 2.84, 95% CI [1.25–6.43]).

**Table 2  Our search strategy of PubMed, the Cochrane Library, Embase, Web of Science and Scopus.** The search time limit is from the establishment of databases to March 19th 2022. Characteristics of the included studies.

| Study | Cancer type | Expression | Detecting method | Sample size | Mean age (range) | Follow-up (month) | Cut-off | High expression (%) | Endpoints |
|---|---|---|---|---|---|---|---|---|---|
| *Feng et al. (2021)* | Esophageal squamous cell carcinoma | Protein | IHC | 127 | 66(43–84) | 168 | staining intensity scores 2-3 | 49 | OS, PFS |
| *Ji et al. (2018)* | Non-small cell lung cancer | Protein | IHC | 254 | 64.8 | 60 | positive cells scored more than 3 | 43 | OS |
| *Kinoshita et al. (2013)* | Hepatocellular carcinoma | mRNA | RT-qPCR | 130 | 66 | 60 | based on expression status | 26 | OS, RFS |
| *Lee et al. (2018)* | Oral cavity squamous cell carcinoma | Protein | IHC | 231 | 57(23–88) | 180 | percentage of immunepositive cells more than 20% | 58 | OS, RFS, |
| *Ma et al. (2017)* | Laryngeal squamous cell carcinoma | Protein | IHC | 327 | 65 | 125 | positive cells scored 3-7 | 54 | OS, RFS |
| *Namikawa et al. (2015)* | Hepatocellular carcinoma | Protein | IHC | 84 | 62(29–81) | 133 | Stained tumour cells score more than 1 | 65 | OS, RFS |
| *Shiozaki et al. (2014)* | Esophageal squamous cell carcinoma | Protein | IHC | 70 | 61.7 | 60 | xCT positivity rate more than 20% | 50 | OS |
| *Sorensen et al. (2018)* | Glioma | Protein | IHC | 215 | 61.9 | 60 | staining intensity scored 1-3 | 21 | OS |
| *Sugano et al. (2015)* | Colorectal cancer | Protein | IHC | 304 | 65.8 | 60 | immunopositive cells more than 25% | 68 | OS |
| *Takeuchi et al. (2013)* | Glioblastoma | Protein | IHC | 40 | 63(21–79) | 150 | 50% | 50 | OS, PFS |
| *Toyoda et al. (2014)* | Tongue cancer | Protein | IHC | 85 | 69(33–92) | 115 | stained tumour cells score more than 3 | 21 | OS, PFS |
| *Zhang et al. (2018a)* and *Zhang et al. (2018b)* | Liver carcinoma | Protein | IHC | 88 | 55(28–76) | 50 | intensity scores more than 6 | 50 | OS |

**Notes.**

IHC, Immunohistochemical; RT-qPCR, Real Time quantitative Polymerase Chain Reaction; OS, Overall Survival; PFS, Progression-free Survival; RFS, Recurrence-free Survival.

## Correlation between SLC7A11 expression and clinicopathological features

To assess the correlation between SLCA711 and clinicopathological features, we combined the extracted OR and exhibited in Table 4. The clinicopathological features includes differentiation (poor *vs* well/moderate), tumor stage (III/IV *vs* I/II), lymph node metastasis (yes *vs* no), lymphatic invasion (yes *vs* no), venous invasion (yes *vs* no) and distant

**Table 3  Result of quality assessment of included studies according to the Newcastle-Ottawa Scale (NOS) for cohort study.**

| Study | Selection | | | | Comparability | | Outcome | | | Total score |
|---|---|---|---|---|---|---|---|---|---|---|
| | S1 | S2 | S3 | S4 | C1 | C2 | O1 | O2 | O3 | |
| *Feng et al. (2021)* | + | + | + | + | + | + | + | | + | 8 |
| *Ji et al. (2018)* | + | + | | + | + | + | + | | + | 7 |
| *Kinoshita et al. (2013)* | + | + | | + | + | + | + | | | 6 |
| *Lee et al. (2018)* | + | + | | + | + | + | + | | | 6 |
| *Ma et al. (2017)* | + | + | + | + | + | + | + | | + | 8 |
| *Namikawa et al. (2015)* | + | + | + | + | + | + | + | | + | 8 |
| *Shiozaki et al. (2014)* | + | + | + | + | + | + | + | | + | 8 |
| *Sorensen et al. (2018)* | + | + | | + | + | + | + | | + | 7 |
| *Sugano et al. (2015)* | + | + | | + | + | + | + | | + | 7 |
| *Takeuchi et al. (2013)* | + | + | + | + | + | + | + | | + | 8 |
| *Toyoda et al. (2014)* | + | + | | + | + | + | + | | + | 7 |
| *Zhang et al. (2018a)* and *Zhang et al. (2018b)* | + | + | | + | + | + | + | | + | 7 |

**Notes.**

+, means that the condition required by Newcastle-Ottawa Scale (NOS) for cohort study is met and is recorded as one point; S1, Representativeness of the exposed cohort; S2, Selection of the non-exposed cohort; S3, Ascertainment of exposure; S4, Demonstration that outcome of interest was not present at start of stud; C1, According to the most important factor to choose control; C2, According to the other important factor to choose control; O1, Assessment of outcome; O2, Follow-up long enough for outcomes to occur; O3, Adequacy of follow-up of cohorts.

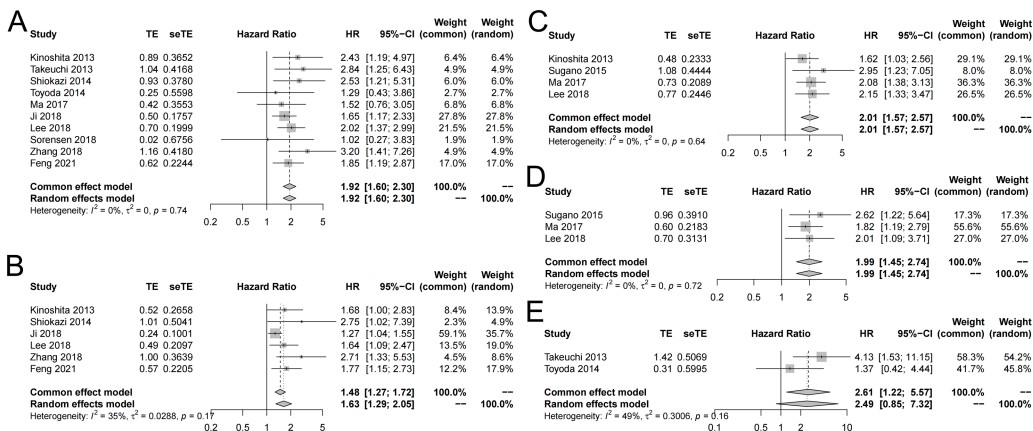

**Figure 2  Forest plot exhibiting the results of meta-analysis.** (A) High expression of SLC7A11 was significantly associated with poor univariate OS. (B) High expression of SLC7A11 was significantly associated with poor multivariate OS. (C) High expression of SLC7A11 was significantly associated with unfavorable univariate RFS. (D) High expression of SLC7A11 was significantly associated with unfavorable multivariate RFS. (E) High expression of SLC7A11 was significantly associated with unfavorable univariate RFS.

metastasis (yes *vs* no). The result indicated that the expression of SLC7A11 was statistically significant associated with tumor stage (OR = 1.77, 95% CI [1.10–2.86], $P = 0.02$, I2 = 61%, random-effect model).

## Cumulative meta-analysis and sensitivity analysis

As shown in the Fig. 3A, with the addition of large samples, the HR values decreased and the 95% confidence interval became narrower. The cumulative meta-analysis reflected the

Peer J

**Table 4    The association between SLC7A11 and clinicopathological features in cancers.**

| Clinicopathologic parameters | Number of studies | Number of cases | Quantitative synthesis | | | Test for heterogeneity | |
|---|---|---|---|---|---|---|---|
| | | | Pooled OR | 95% CI | P value | I² | Model |
| Differentiation (poor vs well/moderate) | 6 | 1147 | 0.75 | 0.55-1.02 | 0.07 | 18% | Fixed |
| Tumor stage (III/IV vs I/II) | 3 | 1005 | 1.77 | 1.10-2.86 | 0.02 | 61% | Random |
| Lymph node metastasis (yes vs no) | 4 | 843 | 1.32 | 0.70-2.49 | 0.39 | 70% | Random |
| Lymphatic invasion (yes vs no) | 4 | 690 | 1.98 | 0.90-4.36 | 0.09 | 61% | Random |
| Venous invasion (yes vs no) | 6 | 920 | 1.40 | 0.74-2.65 | 0.30 | 61% | Random |
| Distant metastasis (yes vs no) | 3 | 750 | 1.40 | 0.93-2.11 | 0.11 | 18% | Random |

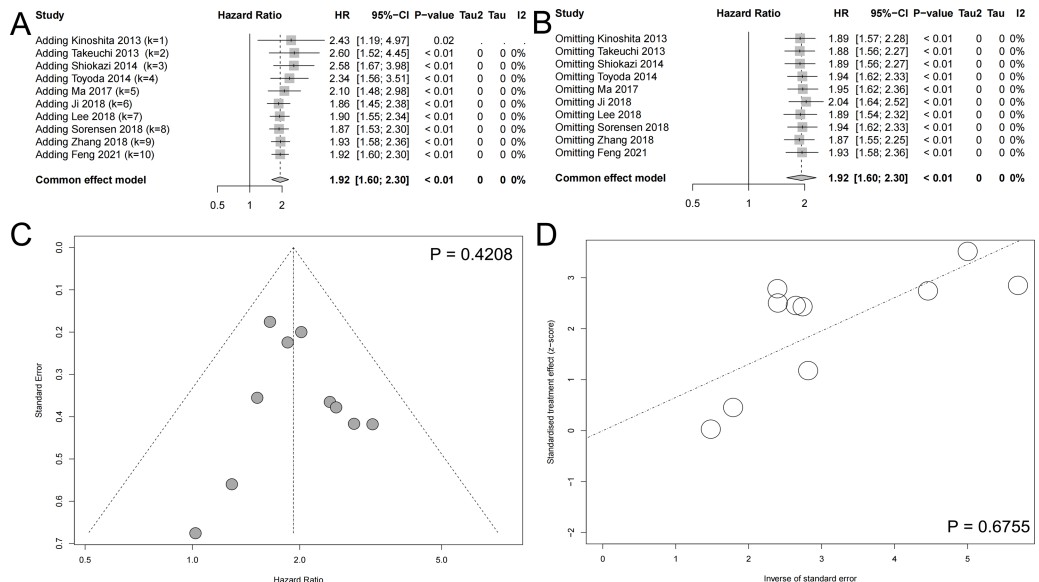

**Figure 3    Other meta-analysis plots.** (A) The cumulative meta-analysis reflected the association between SLC7A11 and unfavorable prognosis more objectively. (B) Sensitivity analysis indicated that the results were robust. (C) The Begg test suggested no significant publication bias. (D) The Egger test suggested no significant publication bias.

association between SLC7A11 and unfavorable prognosis more objectively. Considering the heterogeneity detected in the meta-analysis of OS, sensitivity analysis was applied to explore the stability of meta-analysis. Each included studies reported OS was omitted one by one, in order to detect their influence to the result respectively. The result of sensitivity analysis indicate that our result was robust, which would not statistically significant change when omitting included studies (Fig. 3B).

## Publication bias

In order to detect the publication bias of our systematic review and meta-analysis, The Begg test and Egger test were conducted (*Begg & Mazumdar, 1994*; *Egger et al., 1997*). The Begg test did not reported significant publication and the included studies evenly distributed
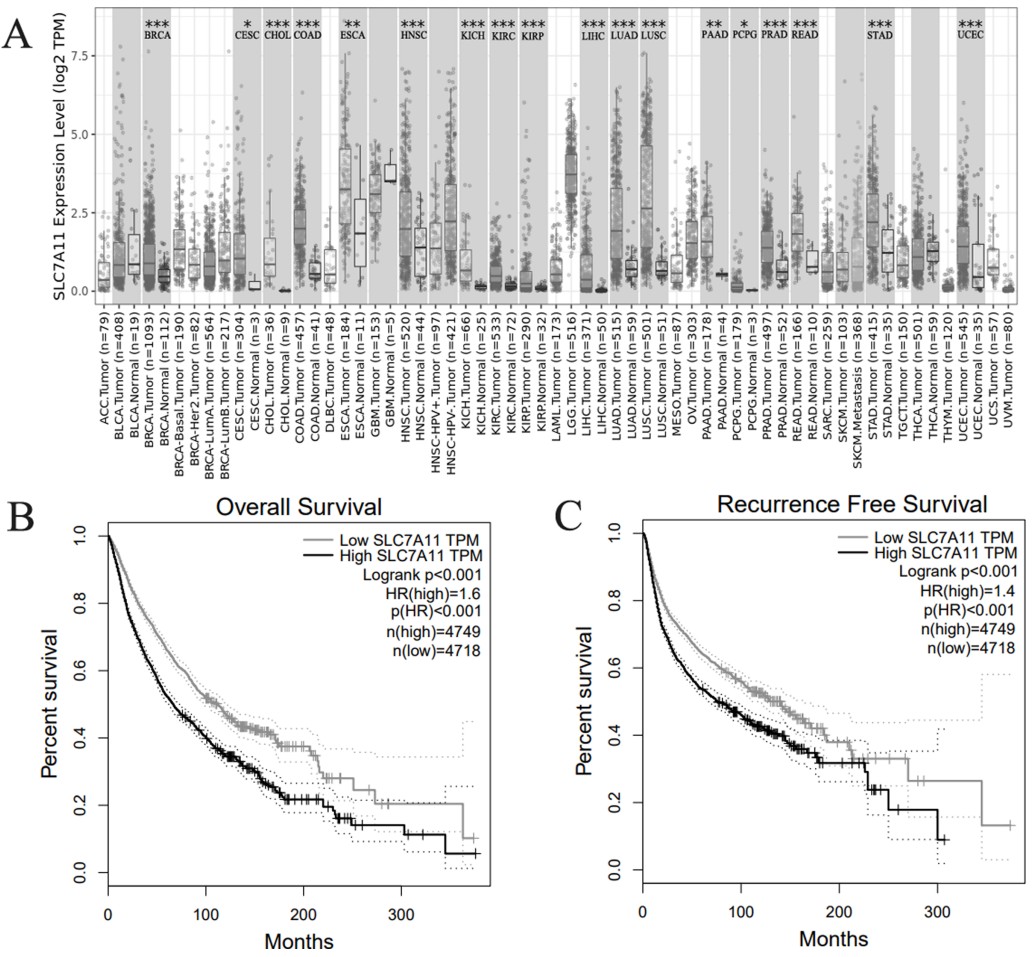

**Figure 4** **Bioinformatics verification in TCGA database.** (A) SLC7A11 was highly expressed in 18 kinds of human cancers. (B) High SLC7A11 expression was associated with poor OS in human cancers. (C) High SLC7A11 expression was associated with poor RFS in human cancers.

on both sides of the midline (Fig. 3C). The Egger test also indicated that there existed on significant publication bias (Fig. 3D).

### Further bioinformatics verification

With the aim to verify our results, we explored the expression and prognosis significance in TCGA database. SLC7A11 expression was significant high in 18 kinds of human cancers (Fig. 4A). Then, we explored the expression of SLC7A11 in TCGA database. We combined all kinds of human cancers together, and the result indicated that high expression of SLC7A11 was significantly associated with unfavorable OS (Fig. 4B) and RFS (Fig. 4C).

## DISCUSSION

In this study, we performed a systematic review and meta-analysis with a total of 1955 patients from 12 studies included. Our results demonstrated that SLC7A11 expression was correlated with more unfavorable prognosis including OS, RFS and PFS. Then, we assessed

the correlation between SLC7A11expresssion and clinicopathological features. The result indicated that high SLC7A11 expression was statistically significant related to advanced tumor stage. In addition, closely correlation between SLC7A11 and advanced stage indicated that tumor progression mediated by SLC7A11 might contribute to SLC7A11 caused worse prognosis. Considering the heterogeneity, we conducted the sensitivity analysis which showed our result was stable and robust. Also, further bioinformatics results further supported our conclusion.

Cancer cells need extracellular nutrient to support their survival and proliferation. Among them amino acids were the most important by supplying building blocks, energy as well as redox hemostasis. Cysteine is a very important amino acid with multiple roles in protein synthesis, post-translational modifications and redox maintenance (*Liu et al., 2021*). Cysteine is the rate-limiting precursor of glutathione, a tripeptide composed of three amino acids (cysteine, glutamate and glycine) that is the most abundant intracellular antioxidant. Intracellular cysteine can be recovered by *ab initio* biosynthesis or by protein degradation, and most cancer cells rely primarily on the cystine transporter protein system Xc- (consisting of the catalytic subunit SLC7A11 and the chaperone subunit SLC3A2) to obtain cystine from the extracellular environment, which is then converted to cysteine in the cytoplasm *via* a reduction reaction that consumes NADPH; cysteine is then used for the synthesis of glutathione (and other biomolecules) (*Liu, Xia & Huang, 2020*). A growing number of studies suggest that SLC7A11-mediated cystine uptake plays a key role in inhibiting oxidative responses and maintaining cell survival under conditions of oxidative stress (*Hémon et al., 2020*). When it comes to its association with the prognosis of cancers, a pan-cancer analysis used survival data from The Cancer Genome Atlas (TCGA) database reached a conclusion which was similar with our systematic review and meta-analysis. Through their bioinformatics approaches, *Lin et al. (2022)* concluded that SLC7A11 was found to be highly expressed in the 20 types of cancer and the up-regulated expression of SLC7A11 was related to poor prognosis in a variety of cancer including, including adrenocortical carcinoma, cervical squamous cell carcinoma and endocardial adenocarcinoma, and so on (*Lin et al., 2022*). The potential mechanism of SLC7A11 in tumor development and progression include resistance to anti-cancer drugs and inhibition of ferroptosis. On the one hand, SLC7A11 overexpression has been shown to correlate with or functionally promote resistance to various anti-cancer drugs, such as cisplatin, gemcitabine, and MAPK pathway inhibitors. SLC7A11 overexpression promotes radio-resistance, whereas SLC7A11 inhibition enhances radio-sensitivity (*Shen et al., 2018*). On the other hand, SLC7A11 is able to promote tumor progression by inhibiting ferroptosis. Studies have shown that loss of tumor suppressors (*e.g.*, p53 and BAP1), mutations in proto-oncogenes (*e.g.*, KRAS) or overexpression of pro-tumor function proteins (*e.g.*, OTUB1) increase the levels of SLC7A11 by upregulating its transcriptional levels or stabilizing its protein, thereby suppressing iron death and promoting tumor development (*Koppula, Zhuang & Gan, 2021*). Studies have found that SLC7A11 is a key target gene of BAP1, and BAP1 inhibits the expression of SLC7A11 by reducing H2Aub on the SLC7A11 promoter, promoting the occurrence of ferroptosis, and then inhibiting tumor growth. Mutations in BAP1 can cause various types of cancer, such as renal cell carcinoma and

mesothelioma, and when BAP1 is mutated, the inhibition of SLC7A11 and the promotion of ferroptosis are lost (*Zhang et al., 2018b*). The results of our systematic review and meta-analysis also indicated that high expression of SLC7A11 was statistically significant related to unfavorable OS, RFS, PFS and clinicopathological features in a number of human cancers. Taking these results together, we could infer that SLC7A11 is a potential oncogene in human cancers.

Normal cells or tissues can compensate for the loss of SLC7A11, cysteine synthesis, or cystine (or cysteine) uptake *via* other transporter proteins by acquiring intracellular cysteine. In contrast, cancer cells are more dependent on SLC7A11-mediated cysteine uptake for cysteine acquisition and maintenance of redox homeostasis than normal tissues. These two features of SLC7A11, its non-essential nature in normal physiology and its high expression in cancer, suggest that targeting SLC7A11 may selectively kill tumor cells and impair tumor growth while preserving normal cells or tissues. Therefore, tumor suppression could be achieved by the following methods: (1) Directly blocking the activity of SLC7A11 cystine transporter protein using various inhibitors. These drugs inhibit SLC7A11 uptake of cystine, which induces lipid peroxidation and iron death. (2) Targeting the glucose dependence of SLC7A11 high expressing cancer cells by inhibiting glucose uptake. Reduction of available glucose in SLC7A11 high expressing cancer cells induces disulfide bond stress, leading to rapid cell death. (3) Targeting glutamine dependence in SLC7A11 high expressing cancer cells by using glutaminase inhibitors (*e.g.*, CB-839) to inhibit cancer cell growth.

However, there also existed some limitations in our study. Firstly, the number of subjects of some included studies was limited. Secondly, the included studies were all retrospective studies. No prospective studies reached the inclusion criteria. Thirdly, the cut-off values of SLC7A11 expression were different among the included studies, which may be a resource of heterogeneity. Therefore, further multi-center prospective studies are still needed.

## CONCLUSION

Our study demonstrated that high SLC7A11 expression is associated with worse prognosis in patients with cancer and more advanced tumor stage. Thus, SLC7A11 expression could be a biomarker for the prognosis of cancers.

### Funding
This study was supported by the Science and Technology Department of Sichuan Province (23ZDYF2384 to Jiantao Wang). The funders had no role in study design, data collection and analysis, decision to publish, or preparation of the manuscript.

### Grant Disclosures
The following grant information was disclosed by the authors:
Science and Technology Department of Sichuan Province: 23ZDYF2384.

## Competing Interests

The authors declare there are no competing interests.

## Author Contributions

- Jiantao Wang conceived and designed the experiments, authored or reviewed drafts of the article, and approved the final draft.
- Siyuan Hao conceived and designed the experiments, performed the experiments, analyzed the data, prepared figures and/or tables, authored or reviewed drafts of the article, and approved the final draft.
- Guojiao Song analyzed the data, authored or reviewed drafts of the article, and approved the final draft.
- Yan Wang analyzed the data, authored or reviewed drafts of the article, and approved the final draft.
- Qiukui Hao analyzed the data, authored or reviewed drafts of the article, and approved the final draft.

## Data Availability

The raw data is available in the Supplemental Files.

## Supplemental Information

Supplemental information for this article can be found online at http://dx.doi.org/10.7717/peerj.14931#supplemental-information.

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
