# Peer review of "The prognostic and clinicopathological significance of SLC7A11 in human cancers: a systematic review and meta-analysis"

_PeerJ, doi:10.7717/peerj.14931_

## Round 0.1 · original submission · Minor Revisions

Please revise the manuscript as the reviewers suggested.

Reviewer 1 ·

Basic reporting

The paper is well written.

Experimental design

Considering the nature of the meta-analysis, I suggest the authors add more details about the study included.
(1) Suggest adding details related to study types for clinical study- neoadjuvant study, adjuvant study, and non-adjuvant study?
(2) If clinical trials were used, were all treatment arms extracted? I suggest user adjust the PFS and OS analysis by treatment or mechanism of action.
(3) RFS: definition should be discussed slightly. Was the censoring rule the same across all studies?
(4) RFS and PFS were pulled together, suggesting separation as RFS is often used in an adjuvant setting while PFS is not.

Validity of the findings

Comparing plot B and plot C, the OS seems lower than RFS. This seems not right as an OS event is counted as part of PFS/RFS events, so one shall always expect lower PFS/RFS than OS. Suggest to double check and examining if it is not a typo.

Additional comments

Minor comments:
L76 - ‘subjected’ should be ‘subjects’?
L90,’methodology published by Tierney’, suggests spelling out the method and adding some details to it.
Figure 2, description, ‘significant’ should be ‘significantly’. Add footnote for ‘se’

Reviewer 2 ·

Basic reporting

In general, the manuscript was well-written using professional and easy-understandable language. The introduction offered a clear overview of the gene SLC7A11. However, the authors ignored the purpose of this article is about finding cancer biomarkers, they failed to provide background information of the current situation of other cancer biomarkers in general. Readers of this article may feel confused about how good or bad SLC7A11 comparing to current other biomarkers for cancer. In the methods part, the authors provided details of the methods. In the results part, the authors reported the correlation between SLC7A11 expression and overall survival and RFS/PFS. However, the authors did not provide the correlation between SLC7A11 and some other biomarkers or closely related genes. Those information can be very useful for readers who are interested in explaining the discovery of the authors mechanistically. The figures are well prepared and with detailed captions. But the layout of the figures makes it a little hard for the readers to subtract information from them because the texts are too small (Fig. 2, 3, 4). Meanwhile, the format of the tables needs to be adjusted as some columns are cut by the edge of the paper (Table 2, 3, 4). References and raw data are provided. Overall, the manuscript used the method of literature search and meta-analysis and provided a well-explained picture of SLC7A11 as a biomarker for clinical prognosis use. However, the manuscript is too limited to gene SLC7A11 itself and lacks a further discussion on other biomarkers and on different cancer types, etc.

Experimental design

The literature search and analysis workflow were easy to understand. The article is a meta-analysis paper without many bench experiments.

Validity of the findings

The finding of the manuscript is novel and can serve as a hint for further cancer prognosis clinically. However, the article failed to compare SLC7A11 as a biomarker comparing other biomarkers for similar use. The author focused too much on gene SLC7A11 itself and lost the big picture of the cancer biomarker field. Further work and discussion are needed for the manuscript.

Additional comments

If supplemented with the mentioned discussion above, this manuscript can be more meaningful to a wider reader population.

---

## Round 0.2 · accepted · Accept

This manuscript can be accepted now.

Reviewer 1 ·

Basic reporting

This paper is well-written in professional English, self-contained and professionally structured.

Experimental design

To perform a systematic review regarding the prognostic effect of SLC7A11 on human cancer, the author performed a meta-analysis using 1955 subjects from 12 studies.

Validity of the findings

The results suggested that SLC7A11 expression led to lower OS, RFS, and PFS. The association between SLC7A11 and tumor stage might explain such prognostic effect of SLC7A11.

Reviewer 2 ·

Basic reporting

The authors addressed most of the points from the last review. They have responded to the questions quite well.
Although the authors mentioned the correction of Table 2, 3, and 4 for the width problem in the response, on my screen, it seems they are still too wide. However, maybe that is the problem with my PC. It would be good if this can be double-checked before publication.

Experimental design

The authors responded to my suggestion well.

Validity of the findings

The authors responded to my suggestion well.